# Herbicide Bioassay Using a Multi-Well Plate and Plant Spectral Image Analysis

**DOI:** 10.3390/s24030919

**Published:** 2024-01-31

**Authors:** Seung-Min Jeong, Tae-Kyeong Noh, Do-Soon Kim

**Affiliations:** Department of Agriculture, Forestry and Bioresources, Research Institute of Agriculture and Life Sciences, College of Agriculture and Life Sciences, Seoul National University, Seoul 08826, Republic of Korea; tmdals619@snu.ac.kr (S.-M.J.); pheno_tk@snu.ac.kr (T.-K.N.)

**Keywords:** herbicide bioassay, mode of action, high-throughput screening, spectral image analysis, multi-well plate assay

## Abstract

A spectral image analysis has the potential to replace traditional approaches for assessing plant responses to different types of stresses, including herbicides, through non-destructive and high-throughput screening (HTS). Therefore, this study was conducted to develop a rapid bioassay method using a multi-well plate and spectral image analysis for the diagnosis of herbicide activity and modes of action. Crabgrass (*Digitaria ciliaris*), as a model weed, was cultivated in multi-well plates and subsequently treated with six herbicides (paraquat, tiafenacil, penoxsulam, isoxaflutole, glufosinate, and glyphosate) with different modes of action when the crabgrass reached the 1-leaf stage, using only a quarter of the recommended dose. To detect the plant’s response to herbicides, plant spectral images were acquired after herbicide treatment using RGB, infrared (IR) thermal, and chlorophyll fluorescence (CF) sensors and analyzed for diagnosing herbicide efficacy and modes of action. A principal component analysis (PCA), using all spectral data, successfully distinguished herbicides and clustered depending on their modes of action. The performed experiments showed that the multi-well plate assay combined with a spectral image analysis can be successfully applied for herbicide bioassays. In addition, the use of spectral image sensors, especially CF images, would facilitate HTS by enabling the rapid observation of herbicide responses at as early as 3 h after herbicide treatment.

## 1. Introduction

There is a growing demand for rapid diagnostic methods to evaluate plant responses to chemicals or environmental conditions in plant biology and agrochemical research. The most common method to diagnose plant responses to chemicals is whole-plant assay in a pot condition [1]. This requires a large space to grow plants in a glasshouse and much effort, including labor, time, and cost. To solve these problems, several rapid methods have been developed such as the petri dish assay [2], growth pouch assay [3], trimmed stem node assay [4], and leaf disc assay [5,6,7,8]. A new approach has also been made to detect plant spectral responses after chemical treatment. Chlorophyll fluorescence (CF) imaging was developed to diagnose plant responses to stress, including herbicides in field conditions, using CF to save time, cost, and labor [8,9,10]. Measuring the plant leaf temperature using an infrared (IR) thermal sensor has also been studied to diagnose plant responses to stress including herbicides [11,12,13].

However, the capacity of current methods to detect plant responses is constrained by their lack of flexibility, repeatability, and effectiveness [9]. The primary obstacles to methods of diagnosis are time and space constraints. To effectively demonstrate and evaluate the responses of plants, it is necessary to have a sufficiently wide space for plant cultivation. Moreover, a significant amount of labor and time is necessary to cultivate enough plants to validate the responses. Furthermore, conducting evaluations in the field with a portable device results in delays and inevitable errors in the plant diagnostic procedure. Therefore, it is crucial to create novel bioassay approaches that can provide a compact and simple approach to assess and diagnose plant responses.

In herbicide discovery and development, a herbicide bioassay is essential for not only detecting and quantifying herbicide efficacy but also diagnosing the herbicide mode of action. A multi-well plate is a widely used tool in cell culture for carrying out large-scale biological investigations [14]. Several studies have exhibited the application of multi-well plates for cultivating plants and assessing their responses to herbicides. Little plants or plant parts, such as leaves, were placed into multi-well plates, and a spectral image analysis was applied to diagnose and evaluate the responses of plants under stress conditions, including herbicides [15,16,17]. These studies demonstrate a good synergy between the two methods; however, combining the use of a spectral image analysis has not been previously investigated.

Therefore, this study was conducted to develop a multi-well plate assay method for herbicide screening. To detect plant responses to herbicides, plant spectral images were acquired after herbicide treatment using RGB, IR thermal, and CF sensors and analyzed for diagnosing herbicide efficacy and mode of action.

## 2. Materials and Methods

### 2.1. Plant Materials and Growing the Plant in Multi-Well Plates

As a model plant to diagnose plant responses to herbicides in multi-well plates, a monocot weed crabgrass, *Digitaria ciliaris*, was selected. Crabgrass seeds were collected in 2021, air-dried, and stored in a cold chamber that was maintained at 4 °C until use. A number of tests were conducted to optimize the growing condition of the weed in multi-well plates, which consisted of 24 wells (4 × 6) with 15.5 mm in diameter (SPL Life Sciences, Pocheon, Republic of Korea). Finally, the seeds of the crabgrass were sown on the multi-well plates at a density of 15 seeds per well^−1^ and incubated in a growth chamber (Hanbaek Science, Bucheon, Republic of Korea) that was maintained at 30 °C/20 °C (day/night) with a 16 h photoperiod until the 1-leaf stage.

### 2.2. Herbicide Treatment

When the crabgrass reached the 1-leaf stage, an herbicide was sprayed using a CO_2_-pressurized, belt-driven sprayer (R&D sprayers, Opelousas, LA, USA) fitted with a flat fan nozzle 8001 (Spraying Systems Co., Glendale Heights, IL, USA) to deliver a spray volume of 300 L ha^−1^. Six herbicides with different modes of action were selected: paraquat (PSI inhibitor), tiafenacil (PPO inhibitor), penoxulam (ALS inhibitor), isoxaflutole (HPPD inhibitor), glufosinate (GS inhibitor), and glyphosate (EPSPS inhibitor) (Table 1). Based on our preliminary test and the growth stage of the crabgrass grown in multi-well plates, a quarter dose (×1/4) of their recommended doses was used.

As each multi-well plate has 24 wells arranged in 6 rows with 4 wells per row-1, only two rows were sprayed with each herbicide by covering the other rows with plastic plates large enough to cover the rows. Following herbicide application, the multi-well plates were placed in the growth chamber that was maintained at 30 °C/20 °C (day/night) with a 16 h photoperiod. All treatments were arranged in a completely randomized design with 8 replications in the growth chamber. Following herbicide treatment, the weed plants were relocated to the growth chamber.

### 2.3. Spectral Image Acquisition and Analysis

RGB, chlorophyll fluorescence (CF), and IR thermal images were acquired after herbicide treatment and analyzed using MATLAB R2023b (The MathWorks Inc., Natick, MA, USA).

#### 2.3.1. RGB Images

RGB images of the crabgrass plants growing in multi-well plates were acquired using a CMOS camera (EOS-600D, Canon, Tokyo, Japan) at 3, 6, 24, 48, 72, and 120 h after herbicide treatment (HAT). The acquired RGB images were then analyzed after white balancing and transforming the RGB image into the Lab color space to segment the crabgrass from the image. Otsu thresholding was then applied in an a* channel in Lab color space to eliminate the background (Figure 1). To measure the visual responses and level of greenness in plants, the modified Normalized Difference Index (mNDI) and Excess Green Index (ExG) were calculated by normalizing the red (R), green (G), and blue (B) values of the crabgrass using the following formulae:(1)$$r=\frac{R}{R+G+B},\;g=\frac{G}{R+G+B},\;b=\frac{B}{R+G+B},\;\mathrm{NDI}=\frac{g-r}{g+r}$$
(2)$$\mathrm{mNDI}=\frac{NDI_{h}}{NDI_{0}}\times 100$$
(3)ExG=2g−r−b
where NDI*_h_* and NDI_0_ indicate the NDI values of the weed plants treated with herbicides and the untreated control, respectively.

#### 2.3.2. CF Images

Chlorophyll fluorescence (CF) images of the crabgrass plants in well plates were acquired using an SNU-KIST 2 imaging system (Seoul National University/KIST, Seoul/Gangneung, Republic of Korea) at 3, 6, 24, 48, 72, and 120 HAT. For each CF image, the pixel intensity of the crabgrass was averaged to derive the CF value after removing the background by Otsu thresholding. The CF image parameters were then calculated using the following formulae:(4)$$F_{v}/F_{m}=\frac{F_{m}-F_{0}}{F_{m}}$$
(5)$$\Phi \mathrm{PSII}=\frac{F_{s}-F_{0}}{F_{s}}$$
(6)$$F_{d}/F_{m}=\frac{F_{m}-F_{s}}{F_{m}}$$
where *F*_0_ is the ground CF at the dark-adapted state, *F_m_* is the CF value at 1 s when a plant emits maximum fluorescence, and *F_s_* is the CF value at 60 s when it comes to a steady state. The CF parameters were then normalized by dividing the values of the crabgrass treated with herbicides by those of the untreated control.

#### 2.3.3. IR Thermal Images

IR thermal images of the crabgrass in multi-well plates were acquired using an A65sc infrared camera (FLIR, Wilsonville, OR, USA) mounted in the growth chamber (Gaooze, Suwon, Republic of Korea) that was maintained at 33 °C at 3, 6, 24, 48, 72, and 120 HAT. The image registration algorithm was employed to extract the leaf temperature of the crabgrass by verifying the plant body using the corresponding RGB image segmented in Section 2.3.1. The temperature difference was subsequently calculated using the leaf temperature data derived from an IR thermal image analysis using the following formula:(7)$$\mathrm{Temperature}\;\mathrm{difference}=T_{h}-T_{0}$$
where *T_h_* and *T*_0_ indicate the leaf temperatures of the weed plants treated with herbicides and the untreated control, respectively.

### 2.4. Statistical Analysis

All plant spectral image data were initially subjected to a two-way analysis of variance (ANOVA) to examine statistical significance in the effects of two factors, herbicides and time, on the spectral responses of crabgrass to the herbicides tested. The normalized spectral parameters were then subjected to a principal component analysis (PCA) to test if the spectral responses can be separated and clustered depending on the herbicide mode of action. Initially, a correlation-based PCA was performed for each time point (3, 6, 24, 48, 72, and 120 HAT) separately and then for pooled data with default settings. All statistical analyses were conducted using R 3.2.3 (R Foundation for Statistical Computing, Vienna, Austria).

## 3. Results

### 3.1. Optimization of Growing Crabgrass in Multi-Well Plates

Considering its fast and even germination and compact plant growth, crabgrass was selected as a model plant for this study. Crabgrass seeds were sown in wells of 24- and 96-well plates, and we checked their seedling growth. The well size of the 96-well plate was too small to grow crabgrass, while that of the 24-well plate was large enough to grow crabgrass. Therefore, the 24-well plate was selected for further study. To optimize seed germination and seedling growth, various media, including water, agarose (0.4%, *w*/*v*), and Murashige and Skoog media, were tested, and it was found that that the medium composed solely of agarose (Inclonebiotech Co., Seongnam, Republic of Korea) was appropriate [18]. To optimize growth conditions in the 24-well plate, a number of seeds per well^−1^ and the temperature from germination to seedling growth were tested, and it was found that 15 seeds per well^−1^ appeared to be appropriate, and the optimum growth conditions in the 24-well plate were 30 °C/20 °C (day/night) with a 16 h photoperiod until the 1-leaf stage. At the 1-leaf stage, the crabgrass had a sufficient canopy size to exhibit spectral responses to the herbicides. Therefore, our results demonstrated that crabgrass can be grown in 24-well plates for herbicide bioassays. Crabgrass seeds at 15 seeds per well^−1^ need to be sown in wells containing 0.8 mL of a 0.4% agar medium and incubated in a growth chamber that is maintained at 30 °C/20 °C (day/night) with a 16 h photoperiod until the 1-leaf stage (5 days). After herbicide treatment, the crabgrass needs to be placed in the growth chamber for image acquisition.

### 3.2. Changes in Spectral Response of Crabgrass to Herbicides

The plant spectral images acquired by different sensors, RGB, CF, and IR thermal sensors, show clear differences depending on the herbicide mode of action and the time after herbicide treatment (Figure 2). Spectral changes by herbicide treatment became apparent as the time after herbicide treatment progressed. The fastest spectral change was observed in the paraquat (PSI) treatment in all spectral images, followed by tiafenacil (PPO) and glufosinate (GS), while the slowest one was observed in penoxsulam (ALS), followed by isoxaflutole (HPPD) and glyphosate (EPSPS). Spectral changes in the RGB images include dehydration, chlorosis, and necrosis, which depend on the herbicide mode of action and the time after herbicide treatment (Figure 2A). In the case of the CF images, the most notable and fast change was observed in the paraquat (PSI) treatment, even at 3 HAT, and a distinctive difference between the herbicide modes of action was observed from 48 HAT (Figure 2B). A relatively slow change but still clear difference between the herbicide modes of action was also observed in the IR thermal images (Figure 2C). The ANOVA of all spectral parameters estimated by a spectral image analysis showed a very high statistical significance for both herbicide and time and their interaction (Table 2). Considering F values, herbicide affected the RGB and CF images more significantly than time, while it affected the IR thermal images less significantly than time. In particular, the difference in F_v_/F_m_ and F_d_/F_m_ between herbicide and time were the greatest, suggesting that these CF parameters may be useful to diagnose herbicide modes of action.

### 3.3. Changes in RGB Spectral Responses to Herbicides

As visual symptoms became evident, the mNDI and ExG values experienced a significant reduction (Figure 3). mNDI and ExG exhibited comparable trends to visual symptoms, with a gradual decrease over time. Paraquat (PSI) showed the fastest and greatest reduction in mNDI and ExG values, followed by tiafenacil and glufosinate. The mNDI and ExG values of paraquat dropped to 53.9% and 73.5% compared to the untreated control, respectively, at 6 HAT. This reduction aligned with the rapid herbicidal symptom of paraquat (PSI) demonstrated in the RGB images (Figure 2A). Crabgrass plants exhibited near-fatal conditions at 24 HAT with paraquat (PSI) and 48 HAT with tiafenacil (PPO). Glufosinate (GS) and glyphosate (EPSPS) started to show color changes at 24 HAT and 72 HAT, respectively, coinciding with the decreases in mNDI and ExG values. Isoxaflutole (HPPD) and penoxsulam (ALS) showed minimal herbicidal activity based on visual observations, and no significant changes were observed in mNDI and ExG after their treatment until 120 HAT. These differential changes in mNDI and ExG after herbicide treatment indicate that mNDI and ExG values can be used to diagnose herbicidal activity and modes of action by a plant image analysis.

### 3.4. Changes in CF Spectral Responses to Herbicides

Three CF parameters, F_v_/F_m_, ΦPSII, and F_d_/F_m_, demonstrated differential changes over time after herbicide treatment depending on the herbicide mode of action (Figure 4). Except penoxsulam, which did not show any herbicidal activity against crabgrass, all the other herbicides, namely paraquat (PSI), glufosinate (GS), tiafenacil (PPO), glyphosate (EPSPS), and isoxaflutole (HPPD), resulted in significantly reduced CF parameters compared to the untreated control. Herbicide treatment primarily resulted in a decrease in all the CF parameters, with variations depending on the herbicide mode of action. Except for penoxsulam (ALS), the CF spectral responses to the herbicides were more immediate and pronounced compared to the other spectral responses such as the RGB and IR thermals. Among the herbicides, paraquat showed the fastest and greatest reduction in all the CF parameters, even at 3 HAT, resulting in a significant reduction in F_v_/F_m_, ΦPSII, and F_d_/F_m_ by up to 45.9%, 56.7%, and 31.8% of the untreated control, respectively, and it reached 0% at 6 HAT. Tiafenacil (PPO) showed the second fastest and greatest significant reduction in all the CF parameters from 24 HAT to almost zero at 72 HAT. Glufosinate (GS) and glyphosate (EPSPS) displayed a sudden decrease in ΦPSII at 24 HAT, but their decreases in F_v_/F_m_ and F_d_/F_m_ were not that great when compared to ΦPSII. Isoxaflutole (HPPD) did not exhibit a significant change in all the CF parameters until 48 HAT, but a sudden decrease in ΦPSII was observed at 120 HAT, where no changes in the RGB and IR thermal spectral responses were observed (Figure 3 and Figure 5).

### 3.5. Changes in IR Thermal Spectral Responses to Herbicides

The temperature difference estimated based on the plant leaf temperature determined by the IR thermal image analysis also showed distinctive differences among the herbicides tested in this study (Figure 5). All tested herbicides exhibited distinct temperature variations compared to the untreated control. An initial increase in temperature difference was observed for paraquat, tiafenacil, glufosinate, and glyphosate, and then a decrease was observed at a certain point, with the plants being dehydrated. As observed in the other spectral responses, paraquat (PSI) showed the greatest change in temperature difference, showing an increase until 6 HAT and then a sudden decrease from 24 HAT. Glufosinate (GS) showed a continuous increase in temperature difference until 24 HAT, while tiafenacil (PPO) and glyphosate (EPSPS) showed an increase until 48 HAT and then a decrease afterwards. Isoxaflutole (HPPD) displayed a significant increase as late as 120 HAT, while penoxsulam (ALS) did not show any noticeable change even at 120 HAT.

### 3.6. Principal Component Analysis

The 2-dimensional PCA of all the spectral image parameters observed at different timings after herbicide treatment showed a gradual clustering of herbicides, with time depending on the herbicide mode of action (Figure 6, Appendix A). Paraquat (PSI) exhibited a noticeable separation from the other herbicides at as early as 3 HAT. At 24 HAT, tiafencial (PPO) became independently clustered from the other herbicides. At 48 HAT, glufosinate (GS) and glyphosate (EPSPS) were independently clustered. Isoxaflutole (HPPD) was the last one separated from the other herbicides at as late as 120 HAT. However, penoxsulam (ALS) failed to be independently clustered, even at 120 HAT. The PCA using the pooled data from 3 HAT to 120 HAT demonstrated a clear clustering of herbicides based on the herbicide mode of action except for penoxsulam (ALS), which was not separated from the untreated control due to its very low efficacy against crabgrass (Figure 7, Appendix A). The 3-dimensional PCA for the spectral image parameters observed for each time point and for the pooled data also showed a similar clustering depending on the herbicide mode of action (Appendix A). The quick-acting herbicides such as paraquat (PSI), tiafenacil (PPO), and glufosinate (GS) required only 2 days after herbicide treatment for a clustering by the herbicide mode of action, while the slow-acting herbicides such as glyphosate (EPSPS) and isoxaflutole (HPPD) required a bit of a longer period after herbicide treatment.

## 4. Discussion

### 4.1. Spectral Image Responses to Herbicides

Plant imaging is becoming a core technique for plant phenotyping and diagnoses in plant and agricultural sciences [19]. One of the benefits of a plant image analysis is that it allows for a non-destructive and high-throughput screening of plant responses [19,20,21,22]. In this study, we developed a multi-well plate assay by combining it with a plant spectral image analysis for the rapid diagnosis of herbicide activity and modes of action. Three different sensors, RGB, CF and IR thermal sensors, were used to acquire spectral responses at different time points after herbicide treatment and showed differential spectral responses depending on the herbicide mode of action and the time when the spectral images were acquired, suggesting a potential diagnosis of herbicide modes of action based on spectral image data. Among the spectral image parameters, those related to CF images seemed to be the most responsive to herbicides, particularly ΦPSII (Figure 4). Other image parameters derived from RGB and IR thermal images were also informative, although they appeared to be relatively slow and less responsive than the CF images (Figure 3 and Figure 5).

An RGB image is most commonly used for analyzing vegetation indices associated with plant growth and canopy development and has a broad range of applications, because it is easily accessible and provides informative and high-resolution results, even with commercially available cameras installed in smartphones [23]. In this study, an RGB color change in response to the herbicides was analyzed by an RGB image analysis, particularly the level of greenness in the leaves, as leaf discoloration such as yellowing and bleaching is the most common symptom after herbicide treatment [24]. The mNDI and ExG values showed significant evidence for changes in leaf color, particularly greenness, and a suppression of plant development resulting from herbicide action (Figure 2A and Figure 3). Moreover, the observed changes varied depending on the herbicide mode of action. The herbicides related to reactive oxygen species (ROS), including paraquat (PSI), tiafenacil (PPO) and glufosinate (GS), exhibited rapid changes in these RGB-based parameters. Glyphosate (EPSPS) showed a rather mild and slow response, while isoxaflutole (HPPD) and penoxsulam (ALS) showed very little changes. These parameters allowed for the explanation of plants’ responses to herbicides depending on the mode of action, suggesting that mNDI and ExG are highly useful to quantify plant damage caused by herbicides and to diagnose herbicide modes of action.

Given that photosynthesis requires plant pigments, antioxidant systems, and proteins linked to several enzymes and carbohydrates, any stress experienced by plants is likely to impact photosynthesis [25,26,27,28]. Herbicides act on distinct targets depending on the mode of action and consequently impact photosynthesis in various ways. In our study, paraquat exhibited the most rapid and significant reduction in the CF parameters within 3 HAT, despite the absence of any apparent visual impact, while no discernible difference in the CF images was observed in the penoxsulam (ALS) treatment (Figure 2B and Figure 4). Other herbicides, such as tiafenacil (PPO), glufosinate (GS), glyphosate (EPSPS), and isoxaflutole (HPPD), exhibited distinct herbicide effects in the CF images and parameters based on their respective modes of action. Among these herbicides, tiafenacil (PPO) was the fastest, followed by glufosinate (GS), glyphosate (EPSPS), and isoxaflutole (HPPD). Interestingly, isoxaflutole (HPPD) showed a sudden decrease in the CF parameters at 120 HAT. Using a CF image analysis, herbicide effects could be detected at much earlier timings compared to other image analyses, and the extent of decrease was quite larger. Overall, the CF image analysis was more discernible in terms of differentiating herbicides depending on their modes of action, suggesting its suitability for the high-throughput screening (HTS) of herbicide molecules.

Leaf temperature is strongly correlated with the overall physiological condition of a plant, particularly the conductance of stomata and the rate of photosynthesis. A symptom of plant stress is the closure of the stomata in leaves, which is accompanied by a reduction in the absorption of carbon dioxide during photosynthesis. This ultimately results in a decrease in transpiration [29], and the temperature of stressed leaves rises due to reduced transpiration. Leaf temperature is a reliable indicator for detecting numerous types of stresses in plants, such as drought, salt stress, and herbicides, that affect photosynthesis [30,31,32,33]. In our study, the IR thermal images revealed that the leaf temperature reached its highest point when the effects of the herbicides and visible damage became evident (Figure 2C and Figure 5). An early initial increase in the leaf temperature and its subsequent decrease at a certain point after herbicide treatment varied with the herbicide mode of action, implying that changes in plant leaf temperature measured by an IR thermal image analysis may also be used to explain herbicide modes of action.

### 4.2. Diagnosis of Herbicide Mode of Action Based on Spectral Response Data

The PCA using all spectral data (RGB, CF, and IR thermal image parameters) successfully differentiated herbicides and clustered them depending on their modes of action (Figure 6 and Figure 7). The results demonstrated a sequential separation of the tested herbicides following the order of spectral responsiveness; paraquat (PSI) was the fastest, followed by tiafenacil (PPO), glufosinate (GS), glyphosate (EPSPS), and isoxaflutole (HPPD). Penoxsulam (ALS), which did not have any herbicidal activity against crabgrass in this study, was located very closely to the untreated control until the end of the measurement. The results obtained by the 3-dimensional PCA also show consistency with those by the 2-dimensional PCA, showing more distinct disparities among herbicide modes of action (Appendix A). Our results thus demonstrate that a multi-well plate assay combined with a plant spectral image analysis can be used to diagnose and investigate herbicide modes of action.

### 4.3. High-Throughput Screening of Herbicides Using Multi-Well Plates Assay Combined with Plant Spectral Image Analysis

The use of multi-well plates in screening plant responses to various stresses has been reported in several previous studies [15,16,17]. In these studies, the CF image analysis was only used to diagnose plant responses. In our study, various plant spectral images, not only CF but also RGB and IR thermal images, were used to investigate plant responses to herbicides. To the best of our knowledge, this study was the first attempt at acquiring and analyzing three different types of spectral images for diagnosing herbicide modes of action using multi-well plates. The multi-well plate assay combined with a spectral image analysis showed a favorable synergy in the whole process of the experiment including plant preparation, image acquisition, and analysis. In comparison with the conventional whole-plant herbicide assay, our method can test many herbicidal chemicals in a limited space and time. Multiple studies using the multi-well plate assay can be simultaneously conducted in a controlled growth chamber or laboratory equipped with an image-acquisition system, enabling automatic herbicide bioassays and high-throughput screening (HTS) in herbicide discovery and development. Our method thus can be flexibly and effectively implemented into various agrochemical and plant studies, including herbicide discovery.

Nevertheless, there is still much room to be improved for HTS using our method. In plant preparation, the automatic preparation of a growth medium and growing plants will save time and effort dramatically. An automatic image-acquiring system that collects continual spectral responses will save our efforts and provide a more detailed spectral understanding to diagnose herbicide modes of action. Furthermore, improving the spraying system to ensure that herbicide molecules are accurately treated in each well will increase the efficiency and speed of the multi-well plate assay in herbicide screening. Recently, deep learning algorithms have become a powerful tool for image recognition and classification and have been used in crop pests and stress diagnoses [34,35]. A further application of deep learning to our multi-well plate assay combined with a plant spectral image analysis may improve the efficiency of discovering new herbicide molecules with new modes of action.

## Figures and Tables

**Figure 1 sensors-24-00919-f001:**
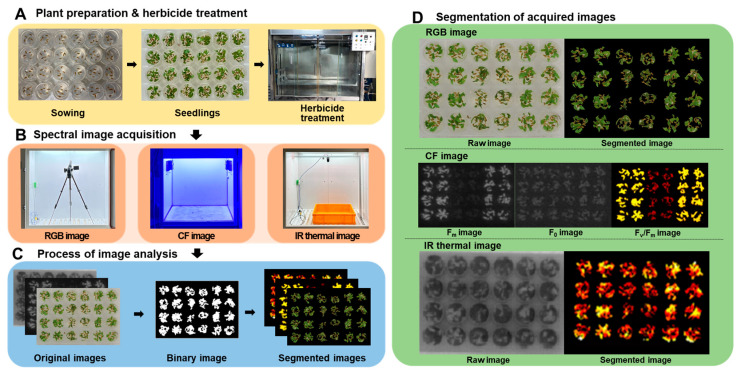
The overall process of multi-well plate assay combined with spectral image analysis. (**A**): Plant preparation and herbicide treatment, (**B**): Spectral image acquisition, (**C**): Process of image analysis, (**D**): Segmentation of acquired images.

**Figure 2 sensors-24-00919-f002:**
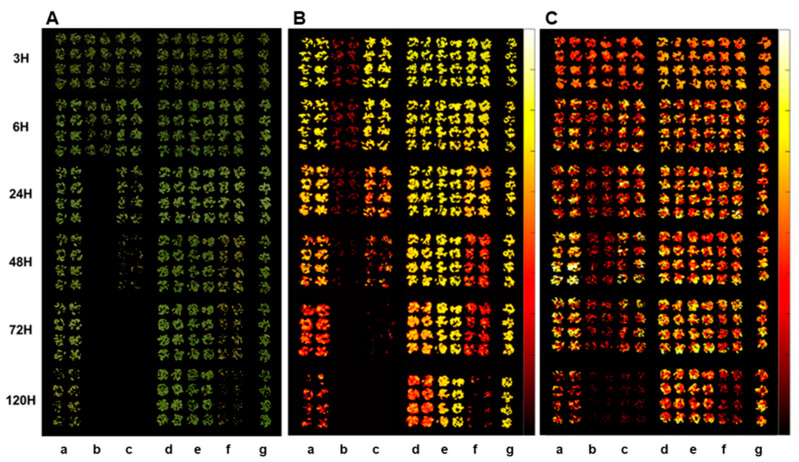
RGB (**A**), CF (**B**), and IR thermal (**C**) images of crabgrass at 3, 6, 24, 48, 72, and 120 h after treatment of herbicides: (a) glyphosate (EPSPS), (b) paraquat (PSI), (c) tiafenacil (PPO), (d) isoxaflutole (HPPD), (e) penoxsulam (ALS), (f) glufosinate (GS), and (g) untreated control.

**Figure 3 sensors-24-00919-f003:**
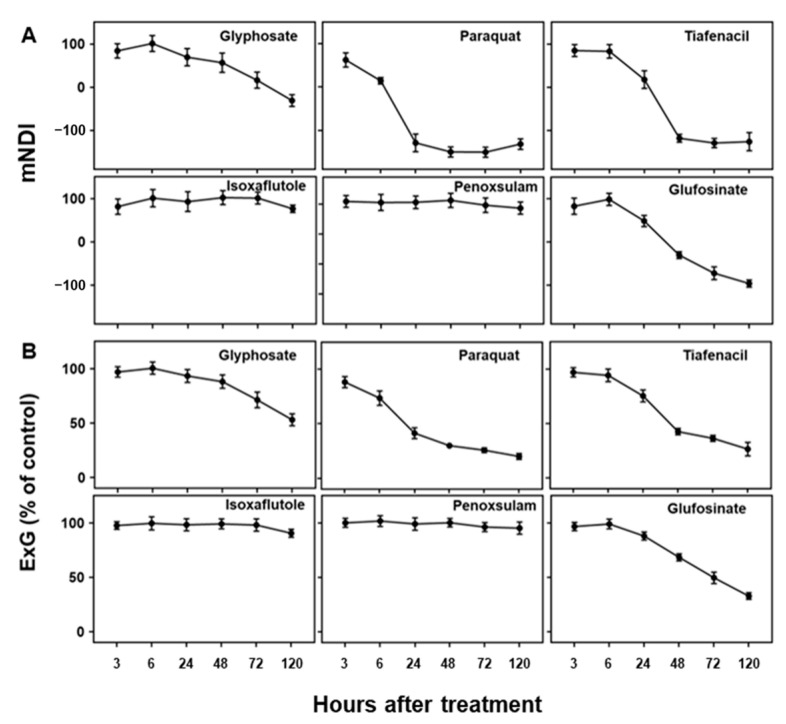
Changes in (**A**) mNDI and (**B**) ExG values over time after treatment of herbicides with different modes of action.

**Figure 4 sensors-24-00919-f004:**
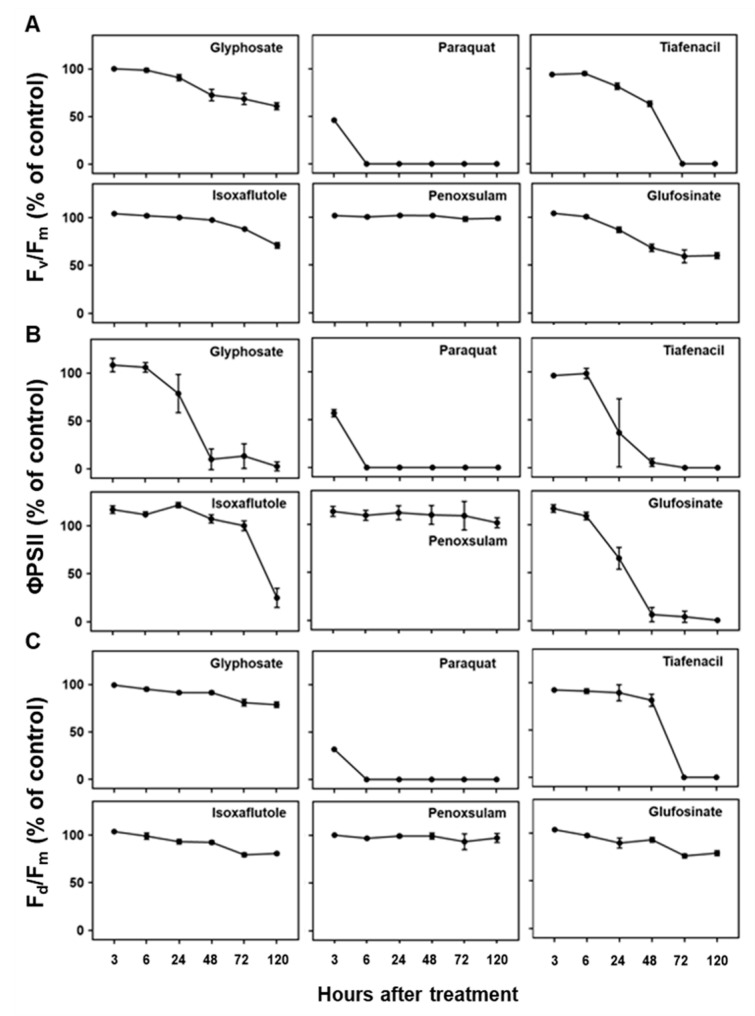
Changes in chlorophyll fluorescence parameters, including (**A**) F_v_/F_m_, (**B**) ΦPSII, and (**C**) F_d_/F_m_, over time after treatment of herbicides with different modes of action.

**Figure 5 sensors-24-00919-f005:**
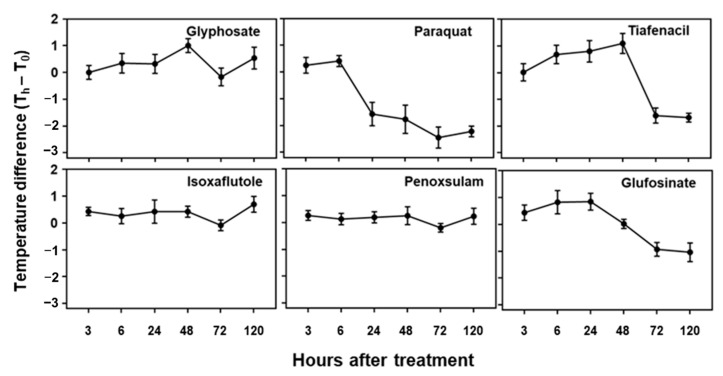
Changes in temperature difference over time after treatment of herbicides with different modes of action.

**Figure 6 sensors-24-00919-f006:**
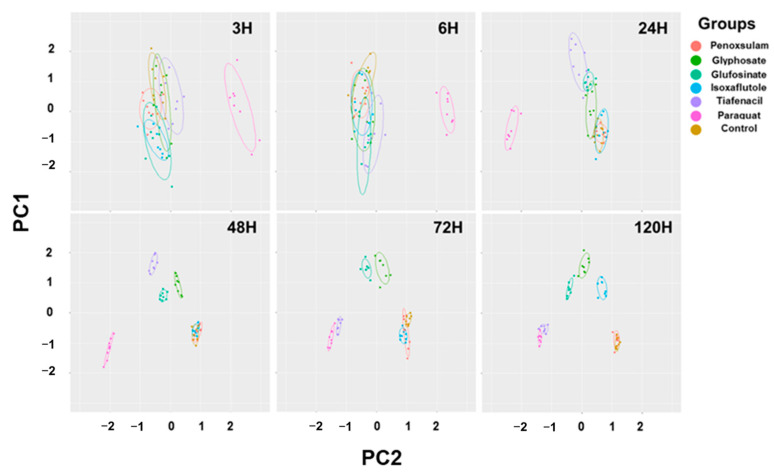
PCA results of six spectral parameters of crabgrass observed at 3, 6, 24, 48, 72, and 120 h after herbicide treatment with different modes of action. Each symbol represents a replication.

**Figure 7 sensors-24-00919-f007:**
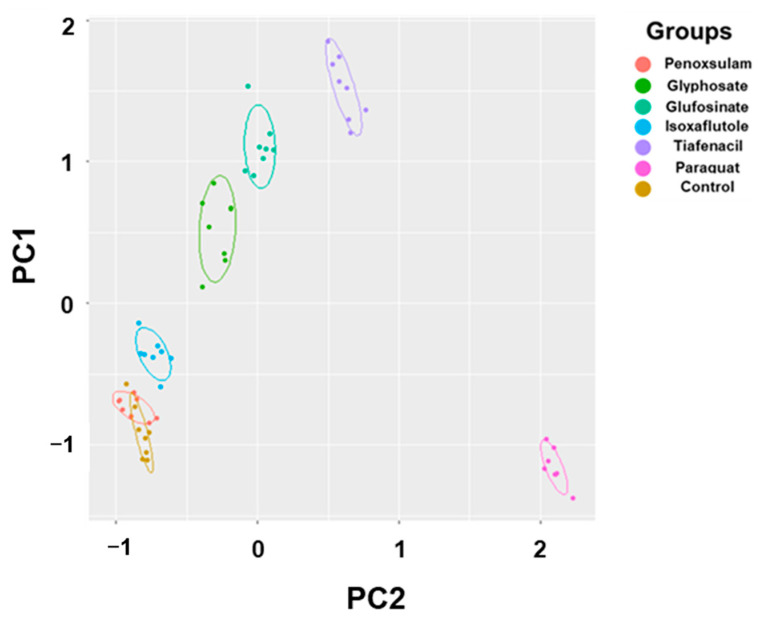
PCA results of six spectral parameters of crabgrass by aggregating all of the data, including 3 h, 6 h, 24 h, 48 h, 72 h, and 120 h after treatment of herbicides with different modes of action. Each symbol represents a replication.

**Table 1 sensors-24-00919-t001:** List of herbicides used for herbicide bioassay using well plates and spectral image analysis.

Herbicide	Mode of Action ^1^	Dose (g a.i. ha^−1^)		Product Name	Formulation ^2^	Manufacturer
		Recommended	Tested			
Paraquat	PSI inhibitor	500	125	Gramoxone	SL	Farmhannong Ltd., Seoul, Republic of Korea
Tiafenacil	PPO inhibitor	160	40	Terrad’or	ME	Farmhannong Ltd., Seoul, Republic of Korea
Penoxsulam	ALS inhibitor	120	30	Salchodaechup	SC	Hankooksamgong Ltd., Seoul, Republic of Korea
Isoxaflutole	HPPD inhibitor	200	50	Merlin	WG	BASF, Lutwigshafen, Germany
Glufosinate	GS inhibitor	1440	360	Basta	SL	Bayer Crop Science Korea, Seoul, Republic of Korea
Glyphosate	EPSPS inhibitor	3690	922.5	Keunsami	SL	Farmhannong Ltd., Seoul, Republic of Korea

^1^ PSI: photosystem I, PPO: protoporphyrinogen oxidase, ALS: acetolactate synthase, HPPD: 4-hydroxyphenylpyruvate dioxygenase, GS: glutamine synthase, EPSPS: 5-enolpyruvylshikimate-3-phosphate. ^2^ SL: soluble concentrate, ME: microemulsion, SC: suspension concentrate, WG: water dispersible granule.

**Table 2 sensors-24-00919-t002:** Summary of two-way ANOVA of each spectral parameter of crabgrass after treatment of herbicides with different modes of action.

Spectral Parameter		Source of Variation	F Value	*p* Value	*p* Value Summary
RGB	mNDI	Herbicide	1063.089	<0.0001	****
		Time	606.495	<0.0001	****
		Herbicide × Time	95.708	<0.0001	****
	ExG	Herbicide	1816.526	<0.0001	****
		Time	1185.414	<0.0001	****
		Herbicide × Time	166.365	<0.0001	****
CF	F_v_/F_m_	Herbicide	8354.295	<0.0001	****
		Time	1894.120	<0.0001	****
		Herbicide × Time	341.633	<0.0001	****
	ΦPSII	Herbicide	859.989	<0.0001	****
		Time	653.637	<0.0001	****
		Herbicide × Time	73.445	<0.0001	****
	F_d_/F_m_	Herbicide	5966.657	<0.0001	****
		Time	661.369	<0.0001	****
		Herbicide × Time	238.078	<0.0001	****
IR	Temperature difference	Herbicide	11.923	<0.0001	****
		Time	37.174	<0.0001	****
		Herbicide × Time	15.847	<0.0001	****

**** represents significance at *p* < 0.0001.

## Data Availability

The data presented in this study are available on request from the corresponding author. The data are not publicly available due to confidentiality.

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
