# Peer review of "Herbicide Bioassay Using a Multi-Well Plate and Plant Spectral Image Analysis"

_sensors, 2024, doi:10.3390/s24030919_

Round 1

Reviewer 1 Report

Comments and Suggestions for Authors

The data analysis methods could be a bit better and perhaps the use of deep learning methods would be more appropriate.

Comments on the Quality of English Language

The English writing of the article is appropriate and it is fluent and simple.

Author Response

We really appreciate your valuable comments on our paper.

Following your comments on deep learning, we modified our discussion to mention a potential application of deep learning to our method. We are planning to apply deep learing method for our future study to improve our method. 

Best regards

Reviewer 2 Report

Comments and Suggestions for Authors

The proposed approach has creativity in contribution and methodology. But, revision in terms of technical details is needed before acceptance. Also, paper organization should be improved. In this respect, some comments are suggested to describe technical details.

1. Did you use any reference to report equation 3 (ExG) or did you propose ExG for the first time? Discuss briefly how this value show green index?

2. What is the meaning of R 3.2.3 in the page 5?

3. Why you didn't compare the performance of your proposed method with existing methods in this scope? Discuss briefly.

4. PCA usually is used for feature reduction. So, it is suggested to discuss about the role of PCA in a more clear way. Also, discuss PCA process with more technical details.

5. The main aim of this paper can be categorized as leaf image analysis problem. You didn’t review deep learning-based methods in this scope completely. So, it is suggested to review more related papers. For example, I find a paper titled “Bark texture classification using improved local ternary patterns and multilayer neural network”, which use combination of bark and leaf images. Cite this paper and some other related.

Author Response

We really appreciate your valuable comments to improve our paper. Following your comments, we revised, please find the resubmitted paper. The below are summary of our revisions.

1. Regarding your first comment on ExG: Excess Green index (ExG) is widely used in remote sensing and image processing to quantify the "greenness" of an area. ExG is particularly useful for highlighting green vegetation in an image and takes advantage of healthy vegetation strongly reflecting green light and absorbing red and blue light. In Discussion, we added a simple comment on greenness

2. R 3.2.3 is the version of the R program and commonly mentioned in papers.

3. We added discussion on a brief comparison between our method and exisitimg ones

4. Regarding PCA, we also added some revision with technical details.

5. Regarding your comment on deep learning, we added some citations of recent studies that utilized deep learning for diagnosing various biotic/abiotic stresses in plants. We also added a potential application of deep learning into our method in the last part of Discussion.

Many thanks again for your valuable comments and kind consideration.

Best regards

Reviewer 3 Report

Comments and Suggestions for Authors

The manuscript presents an original development confirming possibility to distinguish optical properties of plants after their treatment by different herbicides. It demonstrates the authors' extended study with clear demonstration of the obtained results and their grounded interpretation. At whole the work accords to demands of the Sensors journal. However, some revisions/justifications will be reasonable:

1. The title should indicate that the analyzed spectral images are images of plants treated by herbicides.

2. Please list the tested herbicides in the Abstract.

3. Line 42 indicates the lack of flexibility, repeatability and effectiveness for the existing methods registering plant response. However, the improvements reached by the proposed approach are not characterized in these terms. Please give finalizing characterization of the proposed and known techniques by comparison of their corresponding quantitative parameters.

4. The study presents the use of crabgrass that was cultivated at the same conditions with the only difference of herbicide treatment. However, for plants in field many additional factors may influence their statement and cause variation of spectral characteristics. How this variability will be taken into consideration in the course of real field testing? Please comment this issue.

5. Lines 80-82 and table 1. Please specify the used abbreviations. Modes of action should be listed only once: either in text or in table.

6. Section 2.3.3. The choice of 33 C temperature for growing (namely in these experiments) should be grounded.

7. Number of repetitions should be indicated in the legend to Fig. 3 for clarification of the drawn error bars.

8. Concerning Fig. 5. Range of Th values with statistically reliable differences from T0 is not clear for the presented data. Please justify this range to be sure in the comments that are given to this figure (lines 249-257) and comment cases of temperature increase or decrease. How these effects accord to earlier studies of temperature changes of cultivated plants after herbicide treatment?

9. The manuscript discusses the obtained dependences for PC1 and PC2, but initial determination of these parameters is poorly described and needs extension.

10. The comparison with previous studies [15-17] should be more specific. Which approaches in setting up experiments and processing data were common, and which were implemented for the first time?

Author Response

We really appreciate your valuable comments to improve our paper. Following your comments, we revised, please find the resubmitted paper. The below are summary of our responses to your comments.

1 & 2. Regarding your comments on title of our paper and abstract, we revised accordingly. 

3. Regarding your comments on Discussion, we revised based on your comment.

4. Regarding your comment on the use of our method in the field, I fully agree with your concern. As you mentioned, this method cannot  be directly applied to a field study. Actually we developed this method for rapid diagnosis of plant responses to various stresses including herbicide in the controlled envrionment. In herbicide screening for herbicide discovery and early stage of development, most of herbicide studies are conducted in indoor condition (growth chamber, glasshouse). 

5. Regarding abbreviations, we added full names in Table 1.

6. Regarding the temperature for IR thermal image acquisition:  Our previous studies confirmed that air temperature greater than 30 oC showed more discriminating spectral responses than lower temperature. To get better discrimination between herbicides, we used 33 oC in this study.

7. Replication is already mentioned in Materials & Methods

8. Regarding leaf temperature change, our comments on temperature change in response to stress have already made in Discussion (Lines 343-355). We also added on reference (Reference no. 33), which is related to herbicide, which increased leaf temperature. 

9. In our study, principle components were derived by multiplying weights by each variable (6 spectral parameters). The derived PCs are the order of which vector explains the most variability in the dataset consisting of 6 spectral parameters at 6 different timings. Except temperature difference, 5 spectral parameters similarly affected PC1, and PC2 was more dominantly determined by temperature difference. As we used PCA to cluster herbicide mode of action based on spectral image parameters, we didn't mention PCs in detail. 

10. Regarding your comment on the comparison with previous studies [15-17] , we added a brief discussion. 

Many thanks again for your valuable comments and kind  consideration.

Best regards

Round 2

Reviewer 2 Report

Comments and Suggestions for Authors

Most of comments have been considered by authors in this version. The proposed method is described in a more clear way than previous version. New comments are not suggested. 

Reviewer 3 Report

Comments and Suggestions for Authors

The manuscript has been successfully improved and now may be published